# A Novel Synthesis Method of Dumbbell-like (Gd_1−*x*_Tb*_x_*)_2_O(CO_3_)_2_·H_2_O Phosphor for Latent Fingerprint

**DOI:** 10.3390/molecules29163846

**Published:** 2024-08-14

**Authors:** Lei Huang, Jian Qian, Shijian Sun, Zheng Li, Dechuan Li

**Affiliations:** 1School of Physics and Electronic Information, Huaibei Normal University, Huaibei 235000, China; hbnu991210@163.com (L.H.); qianjianwyyx@163.com (J.Q.); sunsj_0105@163.com (S.S.); 2Anhui Province Key Laboratory of Intelligent Computing and Applications, Huaibei Normal University, Huaibei 235000, China

**Keywords:** Gd^3+^, Tb^3+^, fingerprint, phosphor

## Abstract

A novel method for synthesizing dumbbell-shaped (Gd_1−*x*_Tb*_x_*)_2_O(CO_3_)_2_·H_2_O (GOC:*x*Tb^3+^) phosphors using sodium carbonate was investigated. An amount of 1 mmol of stable fluorescent powder can be widely prepared using 3–11 mmol of Na_2_CO_3_ at a pH value of 8.5–10.5 in the reaction solution. The optimal reaction conditions for the phosphors were determined to be 7 mmol for the amount of sodium carbonate and a pH of 9.5 in the solution. Mapping analysis of the elements confirmed uniform distribution of Gd^3+^ and Tb^3+^ elements in GOC:*x*Tb^3+^. The analysis of fluorescence intensity shows that an optimal excitation wavelength of 273 nm is observed when the concentration of Tb^3+^ is between 0.005 and 0.3. The highest emission intensity was observed for GOC:0.05Tb^3+^ with a 57.5% maximum quantum efficiency. The chromaticity coordinates show that the color of GOC:Tb^3+^ is stable and suitable for fluorescence recognition. Latent fingerprint visualization reveals distinctive features like whorls, hooks, and bifurcations. Therefore, the sodium carbonate method offers an effective alternative to traditional urea chemical reaction conditions for preparing GOC:Tb^3+^.

## 1. Introduction

Gd^3+^, as a rare earth ion, exhibits two important properties of the energy level transition and paramagnetism due to its seven unpaired electrons on the 4*f* orbital and high spin relaxation [1,2,3,4,5]. In the biomedical field, the excellent magnetic and optical properties of Gd_2_O(CO_3_)_2_·H_2_O (GOC) have been applied separately in magnetic resonance imaging [6,7] and luminescence [8,9]. In conventional magnetic resonance imaging applications, GOC is only used for magnetic resonance enhancement. When observing fluorescent images of biological tissues, fluorescent dyes like fluorescein isothiocyanate (FITC) and polyethylenimine (PEI) are commonly used [10,11]. A magnetic enhancement reagent with fluorescence can provide higher image sensitivity and resolution for cellular-level microtissue biological imaging, and also reduce the use of fluorescent agents [12,13]. To our knowledge, there are few studies on the luminescence of GOC in current research on magnetic resonance materials, and almost all chemical reactions use urea as the reaction reagent. In chemical reactions, urea can directly react with gadolinium nitrate to form spherical, flower-shaped, and hollow microspheres of GOC [14,15]. Meanwhile, the morphology and particle size of the product can be easily altered by adding glycerol [16], isopropanol [9], diethylene glycol [17], or Na_2_EDTA [18] to the chemical reaction solution of urea.

In the synthesis process, the production of GOC using the urea method demands strict experimental conditions [18]. Factors such as urea dosage, reaction temperature, and reaction time have a significant effect on the final product [19,20]. Changes in reaction temperature and duration can cause the product to transform into Gd_2_(CO_3_)_3_·3H_2_O [21]. If a mixture solution of urea and nitrate solution is heated, the reaction solution will transform into a gel form [22]. These methods that use urea for chemical synthesis are called the urea combustion synthesis method [23,24]. Therefore, the simplification of GOC preparation conditions and research on its fluorescence emission properties will promote GOC to be a bifunctional agent for magnetic resonance imaging and fluorescent imaging.

In this work, a novel method for preparing GOC:*x*Tb^3+^ using sodium carbonate was proposed. The effect of Na_2_CO_3_ dosage and solution pH on the phase structure and fluorescence emission intensity was studied in detail. Meanwhile, the element distribution, optimal doping ratio, optimal excitation wavelength, and quantum efficiency of GOC:Tb^3+^ samples were characterized using scanning electron microscopy and fluorescence spectrometer. Finally, the application of phosphor for fingerprint visualization was explored.

## 2. Results and Discussion

### 2.1. Optimization of Preparation Conditions

Figure 1 illustrates the impact of pH value and Na_2_CO_3_ dosage on the synthesis of GOC:0.05Tb^3+^ when the reaction temperature is 200 °C (8 h). The peak positions of all diffraction peaks are consistent with the Gd_2_O(CO_3_)_2_·H_2_O in the standard card (00-043-0604). In the chemical reactions at a constant pH value of 9.5, the amount of Na_2_CO_3_ has been increased from 3 to 11 mmol, as shown in Figure 1a. The positions of all diffraction peaks in the XRD spectrum remain unchanged; only the intensities change with different Na_2_CO_3_ contents. The sample prepared with 3 mmol of Na_2_CO_3_ exhibited higher intensity on the diffraction peaks at 27 and 39°. When the Na_2_CO_3_ content exceeds 3 mmol, the diffraction intensities of the sample gradually weaken, and the peak positions of the highest diffraction peak change to 21°.

Figure 1b shows the XRD spectra of the samples produced from solutions with different pH values (8.5–10.5) at a constant Na_2_CO_3_ dosage of 7 mmol. The diffraction pattern reveals that the target products GOC:0.05Tb^3+^ can be generated in all samples. The generated samples only differ in the intensity of XRD diffraction peaks. The pH value in the reaction solution is not the key factor in determining the type of product. The most interesting finding is that the dosage of the reaction reagent Na_2_CO_3_ and the pH value of the solution do not alter the final product GOC:0.05Tb^3+^. This is advantageous for the industrial production of the final product in the fields of biological labelling, latent fingerprinting, and nuclear magnetic imaging.

Figure 2 shows the excitation spectra of GOC:0.05Tb^3+^ samples prepared under different reaction conditions. The excitation intensity of the same GOC:0.05Tb^3+^ sample varies with the preparation conditions. The efficient excitation wavelength of the reaction solution changes when the Na_2_CO_3_ content varies. It peaks at 273 nm, followed by 311 nm, with a relatively small variation at 365 nm. In Figure 2a, the sample prepared with 7 mmol Na_2_CO_3_ has the highest excitation intensity at a constant pH of 9.5. Similarly, in Figure 2b, the optimal preparation condition for the sample with a constant Na_2_CO_3_ dosage of 7 mmol is a pH value of 9.5. The difference in Tb^3+^ excitation intensity is mainly due to the spin-allowed 4*f*-5*d* band, which is easily affected by the crystal field [8,25]. When the reaction conditions change, the direction of crystal growth will change accordingly. The change in crystal size will alter the spatial electric field distribution of rare earth ions in the lattice. The results showed that the optimal pH and Na_2_CO_3_ content for sample preparation were 9.5 and 7 mmol, respectively.

### 2.2. Crystal Structures

The X-ray diffraction spectra of the GOC:*x*Tb^3+^ prepared under experimental conditions of 7 times Na_2_CO_3_ and 9.5 (pH) are shown in Figure 3. In the spectrum, the profile of GOC:0Tb is the same as those of GOC:0.5Tb and GOC:1Tb. There are the same positions in the diffraction peaks. Three diffraction spectra are consistent with the standard card (JCPDS:00-043-0604), which indicates that the lattices of GOC:*x*Tb^3+^ can remain stable even when Gd^3+^ ions are replaced by Tb^3+^ ions in a large proportion. The doping concentration of Tb^3+^ ions in GOC prepared by the sodium carbonate method can be increased from 0 to 100%. The doping concentration of Tb^3+^ is much greater than 3% and 5% of the samples prepared by the urea method [8,14,15]. However, the X-ray diffraction spectra of the samples prepared by this method are consistent with those prepared by urea methods [9,15]. In addition, the composition of phosphor highly depends on preparation conditions, such as the concentration of reactants, reaction temperature, and reaction time, in the urea method [14,18]. When using Na_2_CO_3_ as a reagent, it is important to consider the sensitivity of the reaction conditions.

### 2.3. Microstructure

Figure 4 shows the morphologies and energy dispersive spectrum of GOC:*x*Tb^3+^ (*x* = 0–1) prepared at constant reaction conditions (Na_2_CO_3_:7 mmol, pH:9.5). As shown in Figure 4a–g, all of the samples exhibit a uniform dumbbell shape. This shape remains consistent even when the ratios of Tb^3+^ and Gd^3+^ vary, indicating the similar properties and good compatibility of Tb^3+^ and Gd^3+^ ions in the GOC lattice. When the concentration of Tb^3+^ doping is low, the sample exhibits a small dumbbell-shaped structure, approximately 20 micrometers in length, with mushroom-like heads at both ends. An axial grain handle connects the mushroom heads at both ends. With increasing Tb^3+^ doping concentration, more strip-shaped particles aggregate at both ends, and the volume of the mushroom head gradually increases, while the distance between the two mushroom heads becomes shorter. When the concentration of Tb^3+^ ions is greater than 0.7, the two ends of some samples are connected to each other, but the interface boundary is visible. Figure 4h provides an enlarged image of GOC:1Tb. The smaller needle-shaped grains at the end face gradually grow into flakes, and the hemispherical aggregates of the two end faces contact each other to form a large circular aggregate with a diameter of about 40 μm. Figure 4i presents an energy dispersive spectrum of the GOC:0.05Tb^3+^ sample. The energy spectrum analysis results show that the sample contains four elements: Gd, Tb, C, and O, with relative atomic ratios of 28.7%, 1.7%, 21.9%, and 47.7%, respectively. The ratio of Tb content to the total Tb and Gd content is 0.056, which is close to the original stoichiometric ratio of 0.05.

Figure 5 shows the morphological evolution process of the sample. The agglomeration process of the sample is depicted in Figure 5a–c. The agglomeration process can be described as follows. Na_2_CO_3_ is dissolved in deionized water, resulting in a high concentration of CO_3_^2−^ ions in the solution. Subsequently, when the rare earth ion solution is added dropwise to the sodium carbonate solution, the rare earth cations Gd^3+^ and Tb^3+^ combine with the anion CO_3_^2−^ due to the attraction of positive and negative charges, forming Gd_2_O(CO_3_)_2_·H_2_O nanocrystals [9,26]. The grain has active ends that can adsorb more OH^¯^ to achieve autonomous assembly [27]. The growth rate is significantly higher at the ends of the particle compared to the middle. Finally, dumbbell-shaped GOCs are formed in alkaline hydrothermal environments.

### 2.4. Luminescence Spectra

Figure 6 shows the excitation spectra of GOC:*x*Tb^3+^ (*x* = 0–1) monitored at 541 nm. Four intense peaks are located at 273, 311, 347, and 365 nm. The maximum excitation intensity is at 273 nm, followed by 311 and 365 nm. However, two intensity values of excitation light do not always increase with increasing Tb^3+^ concentration, except for 365 nm. From the inserted excitation intensity curve in Figure 6, it can be seen that the excitation peak intensities of 273 and 311 nm increase with the Tb^3+^ doping concentration in the range of 0.001–0.05. When the concentration of Tb^3+^ ions exceeds 0.05, the intensities of excitation luminescence gradually decrease. The variation curves of these two types of excitation intensities indicate the existence of two types of luminescence mechanisms in the transfer process of excited state electrons. According to the excitation spectra, the excitation peaks at 245, 253, 273, 306, and 311 nm originate from the energy transitions of Gd^3+^ from ^8^S_7/2_ ground state to ^6^D_3/2_, ^6^D_9/2_, ^6^I_17/2_, ^6^P_5/2_, and ^6^P_7/2_ excited states [28]. When a sample contains only one rare earth ion, Tb^3+^, all excitation peaks on the excitation spectrum belong to Tb^3+^ in the GOC:*x*Tb^3+^ (*x* = 1) sample. The excitation peaks in the range of 300–380 nm are attributed to the f–f transition of Tb^3+^ [29]. In the comparison of emission peak positions, the 235–270 nm broadband excitation wavelength of Tb^3+^ overlaps with the 245 and 253 nm of Gd^3+^. In addition, another overlapping wavelength is located at 273 nm when comparing the excitation spectra of samples GOC:1Tb^3+^ and GOC:0.05Tb^3+^. The same energy level provides an optional channel for the transmission of excited state electrons by resonance [30]. The band overlap between Gd^3+^ and Tb^3+^ in the Gd_2_O(CO_3_)_2_ host is the same as that in KGd(CO_3_)_2_ [28]. The results indicate that the excited state electrons of Tb^3+^ are generated through two processes: stimulated transitions and resonant energy transfer, when it is excited by the 273 nm wavelength. For Tb^3+^ doping levels between 0.005 and 0.3, excitation at 273 and 311 nm is effective. This is primarily attributed to the high content of Gd^3+^ in GOC:*x*Tb^3+^, where Gd^3+^ ions transfer excited state photons to Tb^3+^ through energy resonance. When the Tb^3+^ content exceeds 0.3, the intensity of f-f transitions of Tb^3+^ ions is proportional to the Tb^3+^ content, and excitation at 365 nm becomes more effective.

Figure 7 shows the emission spectra of GOC:*x*Tb^3+^ under different light excitations. In Figure 7a, it can be seen that under the excitation of 365 nm ultraviolet light, the emission intensity of GOC:Tb^3+^ increases with increasing Tb^3+^ ion concentration. The emission of Tb ions comes from the electronic transition between the ^5^D_4_ energy state and the ground state [31]. As the Tb^3+^ ion content increases from 0.001 to 1, the emission intensity exhibits no concentration quenching. When the Tb^3+^ doping amount is 100%, the emission intensity of GOC:1Tb^3+^ is the highest. In Figure 7b, it can be observed that when 273 nm is used as the excitation wavelength, the luminescence intensities of GOC:*x*Tb^3+^ are relatively high when the Tb^3+^ ion content is in the range of 0.005–0.1, with GOC:0.05Tb^3+^ showing the highest emission intensity. When the Tb^3+^ content exceeds 0.05, the luminescence intensity of GOC:*x*Tb^3+^ begins to decrease, showing concentration quenching. However, as the sample can still emit light under 365nm excitation, it indicates that this concentration quenching is not due to a reduction in ionic spacing, leading to enhanced non-radiative transitions. Therefore, this concentration quenching can be attributed to the enhancement of cross-relaxation between adjacent ions due to the increased concentration. The type of cross-relaxation can be calculated by the Dexter theory with the following formula [32]:(1)$$\frac{I}{x}=K{\left[1+\beta x^{\frac{Q}{3}}\right]}^{-1},$$
where *I* is the emission intensity, *x* is the concentration of Tb^3+^ ions in GOC:*x*Tb^3+^, K and β are constants, and *Q* of 6, 8, and 10 correspond to electric dipole–dipole, electric dipole–quadrupole and electric quadrupole–quadrupole interactions, respectively. The calculated results of GOC:*x*Tb^3+^ (*x* = 0.05, 0.1, 0.3, 0.5, 0.7, 1) are shown in Figure 8; the *Q* value is 6.22, which is closer to 6. Therefore, the concentration quenching of the Tb^3+^ ions is attributed to the electric dipole–electric dipole interaction.

### 2.5. Decay Curve

The fluorescence decay curve is the variation in the emission intensity of the excited state electrons over time, which can characterize the emission process of the excited state electrons. Figure 9 illustrates the fluorescence decay curve for a typical GOC:0.05Tb^3+^ sample under 273 nm excitation with 541 nm as the monitoring wavelength. As can be seen from the figure, the experimental data can be approximated well with the double exponential equation [1].
(2)$$I\left(t\right)=A_{1}\exp\left(-\frac{t}{\tau_{1}}\right)+A_{2}\exp\left(-\frac{t}{\tau_{2}}\right),$$
(3)$$\tau =\frac{A_{1}\tau_{1}^{2}+A_{2}\tau_{2}^{2}}{A_{1}\tau_{1}+A_{2}\tau_{2}},$$
where *I* is the emission intensity of GOC:*x*Tb^3+^, A is a constant, *t* is time, and *τ* is the ^5^D_4_ energy level lifetime of Tb^3+^.

According to the decay curve, the lifetime of the GOC:*x*Tb^3+^ sample is shown in the inset of Figure 9. The lifetime does not change significantly before the Tb^3+^ doping concentration is less than 0.05, and then decreases rapidly. The lifetime of GOC:0.05Tb^3+^ is 1.4 ms, which is the same as the lifetime of GOC:0.05Tb^3+^ samples prepared by the urea method [9]. The change in the lifetime of the GOC:*x*Tb^3+^ is due to energy transfer between Gd^3+^-Tb^3+^ and Tb^3+^-Tb^3+^. When the concentration of Tb^3+^ is very low, there are many Gd^3+^ ions and their ^6^I_17/2_ energy level can absorb the excitation photons of 273 nm as much as possible, and then transfer the excited state electrons to the ^5^F_3_ energy level of Tb^3+^ [28]. The number of electrons on the excited state energy level of Tb^3+^ tends to saturate, and there is very little change in the lifetime. There are two reasons for the decrease in fluorescence lifetime. Firstly, the concentration of Gd^3+^ ions in the GOC:*x*Tb^3+^ sample decreases, which weakens the energy transfer effect of Gd^3+^ on Tb^3+^. Secondly, as the concentration of Tb^3+^ increases, the number of electrons in the ^5^D_4_ excited state is consumed by the electrical dipole–dipole interaction between Tb^3+^ and Tb^3+^, which weakens the emission intensity of Tb^3+^.

Table 1 shows the quantum efficiency of GOC:*x*Tb^3+^ under excitation at wavelengths of 273, 311, and 365 nm. From the table, it can be seen that the trend of quantum efficiency changes under 273 and 311 nm excitation is the same. However, the quantum efficiency corresponding to 365 nm excitation light increases continuously with increasing Tb^3+^ ion concentration. The maximum quantum efficiency values of the sample under 273 and 311 nm light excitation are relatively high, while the quantum efficiency under 365 nm excitation is relatively low. The maximum quantum efficiency under 273 nm excitation is 57.50%, which is 70.47% higher than that under 365 nm excitation. The quantum efficiency of this system is close to 45% of Gd_2_O_3_:Tb^3+^ and 47% of Gd_2_O_2_S:Tb^3+^ [33,34]. From the above analysis, it can be concluded that high-intensity green light emission can be achieved at lower Tb^3+^ concentrations using 273 nm ultraviolet light in the GOC:*x*Tb^3+^ system.

### 2.6. Chromaticity Coordinates

In the GOC:*x*Tb^3+^ system, the luminescent ions are Gd^3+^ and Tb^3+^. When 273 nm is selected as the excitation wavelength, the emission peak of Gd^3+^ ions is located at 311 nm in the ultraviolet region, corresponding to the energy level transition between ^6^P_7/2_ and ^8^S_7/2_ [28]. Meanwhile, 365 nm is also the effective excitation wavelength of phosphor. The emission of Tb^3+^ has four bands located at positions 486, 541, 588, and 618 nm, all of which are in the visible spectrum region, corresponding to energy level transitions from ^5^D_4_ to ^7^F_6_, ^7^F_5_, ^7^F_4_, and ^7^F_3_ [35]. Figure 10 illustrates the chromaticity coordinates of GOC:*x*Tb^3+^ (*x* = 0.001–1) in the 420–700 nm wavelength range under 273 nm excitation. From the graph, it can be observed that the chromaticity coordinates of GOC:*x*Tb^3+^ are close to (0.33, 0.58) when the concentration of Tb^3+^ is varied from 0.001 to 1. The yellow-green light emitted by Tb^3+^ in GOC:*x*Tb^3+^ remains stable, showing consistent emission characteristics with varying Tb^3+^ ion concentrations. The stable emission characteristics of GOC:*x*Tb^3+^ samples can be used for applications such as bioluminescence and anti-counterfeiting.

### 2.7. Latent Fingerprint

To evaluate the imaging quality of latent fingerprints on glass, a common 365 nm ultraviolet excitation light is used for fingerprint visualization. Figure 11 gives a latent fingerprint pattern made using GOC:1Tb^3+^ phosphor. As can be seen from the figure, the excited phosphor clearly characterizes the primary and secondary structures of the fingerprint. In the primary structure, the shape of the fingerprint is a closed ellipse, which belongs to the whorl structure, and is marked as 1. In the secondary structure, the characteristic details of the fingerprint, such as the ridge ends, islands, hooks, crosses and bifurcations, and other features, can be observed, and are marked as 2, 3, 4, 5, and 6, respectively. In addition, the characteristic ridges and furrows of the fingerprints can be represented directly by using the grey value change curves on a specific straight line (marker 7). The gray curve of the fingerprint exhibits five distinctive ridge peaks, which correspond to the five fingerprints present in the selected region. The peaks and valleys observed in the curve correspond to the ridges and furrows, respectively. These results demonstrate that the UV-excited GOC:Tb^3+^ phosphor can be employed for fingerprint detection.

## 3. Materials and Methods

(Gd_1−*x*_Tb*_x_*)_2_O(CO_3_)_2_·H_2_O (x = 0–1) phosphors were prepared by the sodium carbonate method. The raw materials for the chemical reaction were rare earth terbium nitrate, gadolinium nitrate, and sodium carbonate, and all reagents were purchased from Shanghai Aladdin Biotechnology Co., Ltd. (Shanghai, China). In the experimental procedure, amounts of Gd(NO_3_)_3_·6H_2_O (99.99%) and Tb(NO_3_)_3_·6H_2_O (99.99%) were weighed according to the stoichiometric ratio of 1 mmol GOC:Tb^3+^. Secondly, two nitrates were dissolved in 3 mL of water to prepare a mixed nitrate solution. Then, different amounts of Na_2_CO_3_ were weighed and dissolved in 25 mL of water to prepare a sodium carbonate aqueous solution. Finally, the nitrate solution was added dropwise to the vigorously stirred sodium carbonate solution, and the pH value of the mixed solution was adjusted with dilute nitric acid. After vigorous stirring for 20 min, the mixed solution was transferred to a hydrothermal reactor, heated to 200 °C, and kept at that temperature for 8 h. After the reaction was complete, the precipitate was washed and dried to obtain GOC:*x*Tb^3+^ fluorescent powder.

Fluorescent powder is used to visualize latent fingerprints. Firstly, volunteers wash their hands with soap, then lightly touch their faces to leave their fingerprints on the glass. Secondly, the ground Tb^3+^-doped GOC phosphor is sprinkled on the fingerprint, and the glass slide is gently shaken to remove excess powder. Repeat the same steps several times until the fingerprint ridges are filled with fluorescent powder. Under 365 nm ultraviolet irradiation, fingerprint excitation images were captured and recorded by a mobile phone (Honor 30).

The phase structure of the GOC:*x*Tb^3+^ sample was analyzed by X-ray diffraction in the range of 5–80° (PANalytic, Almelo, The Netherlands). The microstructure and elemental composition were examined via scanning electron microscopy (Regulus 8220, Hitachi High Tech Co., Tokyo, Japan). The fluorescence characteristics, lifetime, and quantum efficiency of GOC:*x*Tb^3+^ were measured using FLS920 fluorescence spectrometer (Edinburgh Instruments, Livingston, UK). The gray values of the fingerprints were identified using ImageJ software (V1.8.0.345).

## 4. Conclusions

A novel sodium carbonate method was used to prepare GOC:*x*Tb^3+^ fluorescent powder. This method greatly reduces the dependence of the products on the chemical reaction conditions and improves the strict preparation conditions of the urea method. XRD test results show that the reaction products are all pure phase GOC:*x*Tb^3+^ when the amount of Na_2_CO_3_ is increased from 3 mmol to 11 mmol, and the pH value of the reaction solution can be varied from 8.5 to 10.5 during the chemical reaction. The maximum emission intensity of the samples is obtained at an amount of 7 mmol and a pH value of 9.5 for Na_2_CO_3_. When 273 nm was used as the excitation light source, 5% Tb^3+^ could achieve the maximum intensity with a maximum quantum efficiency of 57.50%. When the concentration of Tb^3+^ ions is higher, the electrical dipole–dipole coupling between Tb^3+^ ions and the decrease in Gd^3+^-Tb^3+^ energy transfer efficiency will reduce the excitation efficiency of 273 nm. GOC:0.05Tb^3+^ phosphor can be used to produce fluorescent fingerprints, which can clearly extract the unique secondary microstructure of the fingerprint. The results indicate that the use of the new carbonate method can more conveniently prepare efficient fluorescent luminescent materials, providing a new way for the mass production of GOC:Tb^3+^ phosphors.

## Figures and Tables

**Figure 1 molecules-29-03846-f001:**
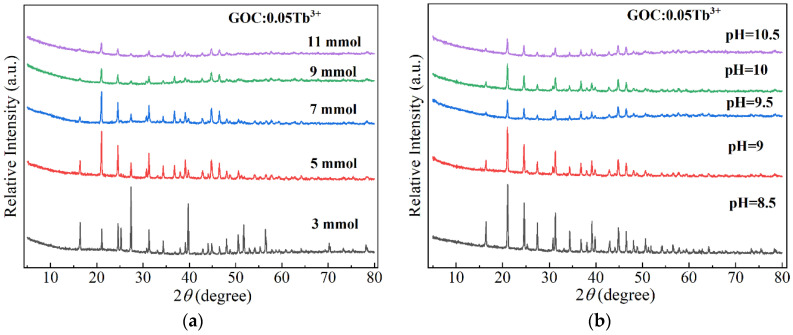
XRD patterns of GOC:0.05Tb^3+^ prepared at different (**a**) contents of Na_2_CO_3_ at a constant pH value of 9.5; (**b**) pH values at a constant Na_2_CO_3_ dosage of 7 mmol.

**Figure 2 molecules-29-03846-f002:**
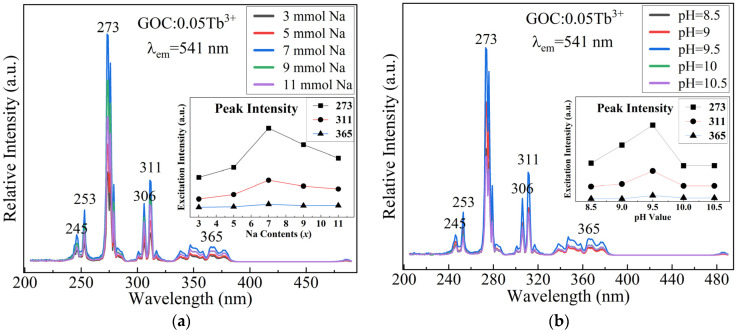
Excitation spectra of GOC:0.05Tb^3+^ prepared at different (**a**) contents of Na_2_CO_3_ (pH:9.5); (**b**) pH values of the reaction solution (Na_2_CO_3_:7 mmol).

**Figure 3 molecules-29-03846-f003:**
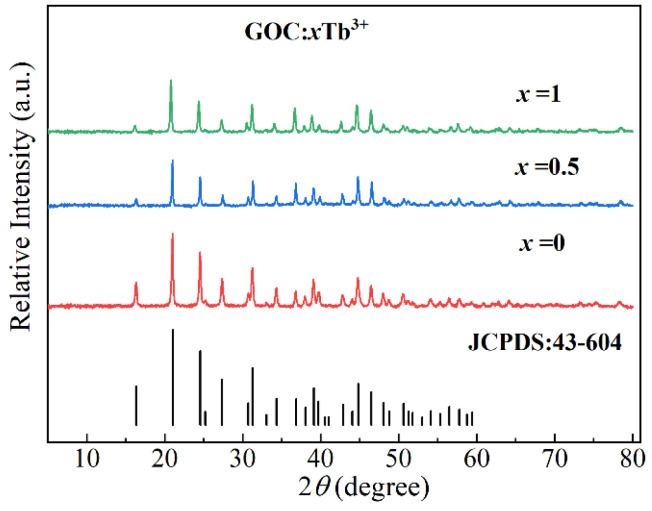
XRD patterns of (Gd_1−*x*_Tb*_x_*)_2_O(CO_3_)_2_·H_2_O.

**Figure 4 molecules-29-03846-f004:**
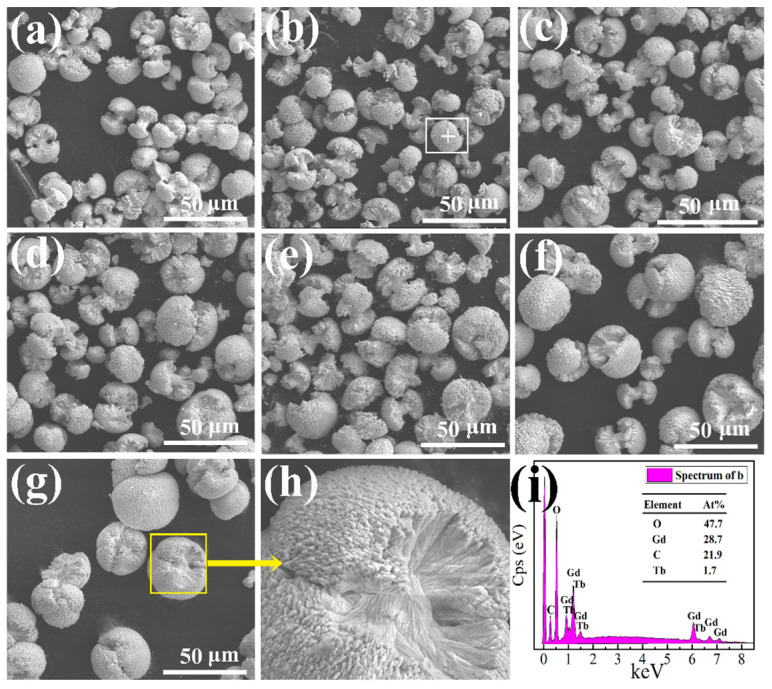
(**a**–**g**) Morphologies of GOC:*x*Tb^3+^ (*x* = 0, 0.05, 0.1, 0.3, 0.5, 0.7, 1); (**h**) enlarged image of GOC:1Tb^3+^ sample; (**i**) energy dispersive spectrum of GOC:0.05Tb^3+^ sample.

**Figure 5 molecules-29-03846-f005:**
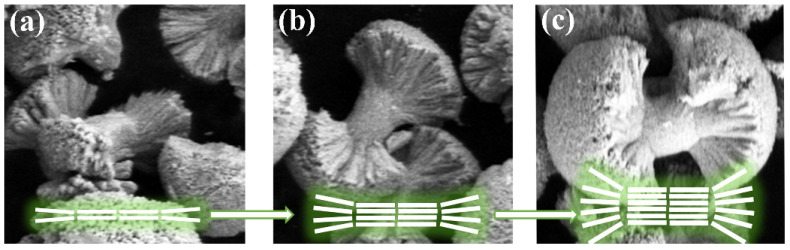
Schematic illustration for the formation process of the dumbbell-like GOC:*x*Tb^3+^ (**a**) aggregate; (**b**) grow; (**c**) form.

**Figure 6 molecules-29-03846-f006:**
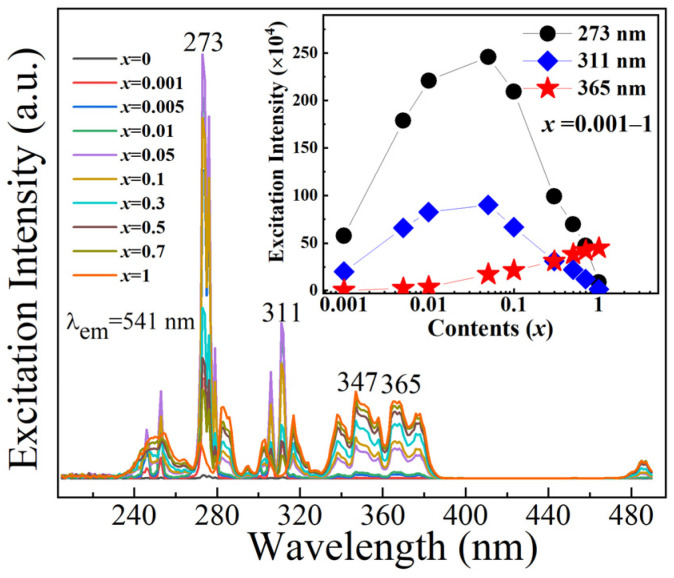
Excitation spectra of GOC:*x*Tb^3+^ (*x* = 0–1).

**Figure 7 molecules-29-03846-f007:**
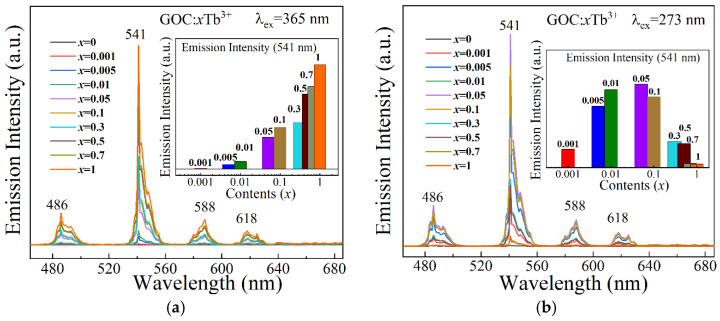
Emission spectra of GOC:*x*Tb^3+^ excitation at (**a**) 365 nm; (**b**) 273 nm.

**Figure 8 molecules-29-03846-f008:**
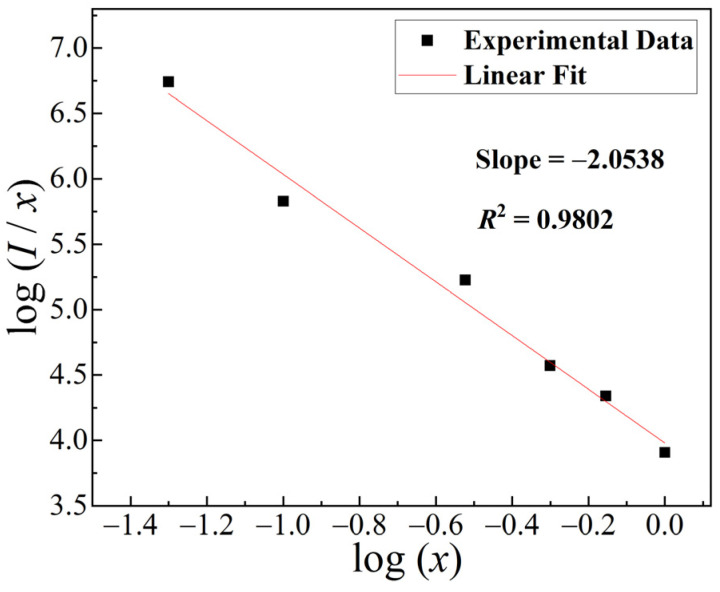
Plot of log (*I*/*x*) as function of log(*x*) in GOC:*x*Tb^3+^ (*x* = 0.05–1).

**Figure 9 molecules-29-03846-f009:**
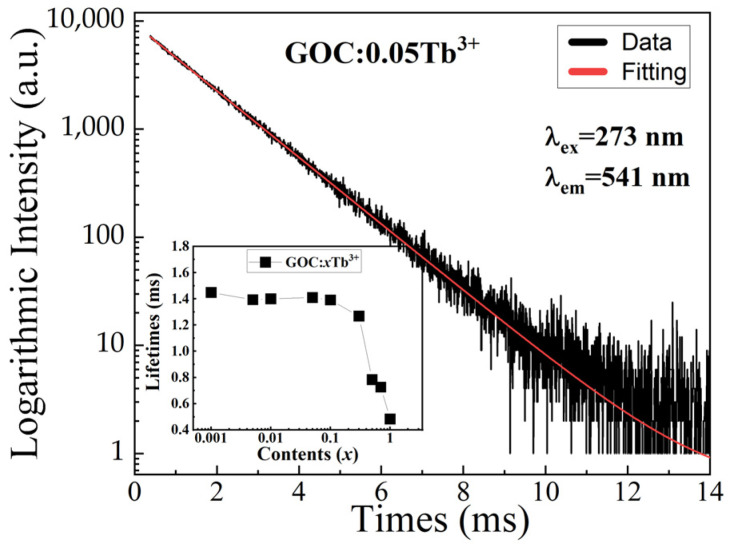
Typical decay curve of GOC:0.05Tb^3+^.

**Figure 10 molecules-29-03846-f010:**
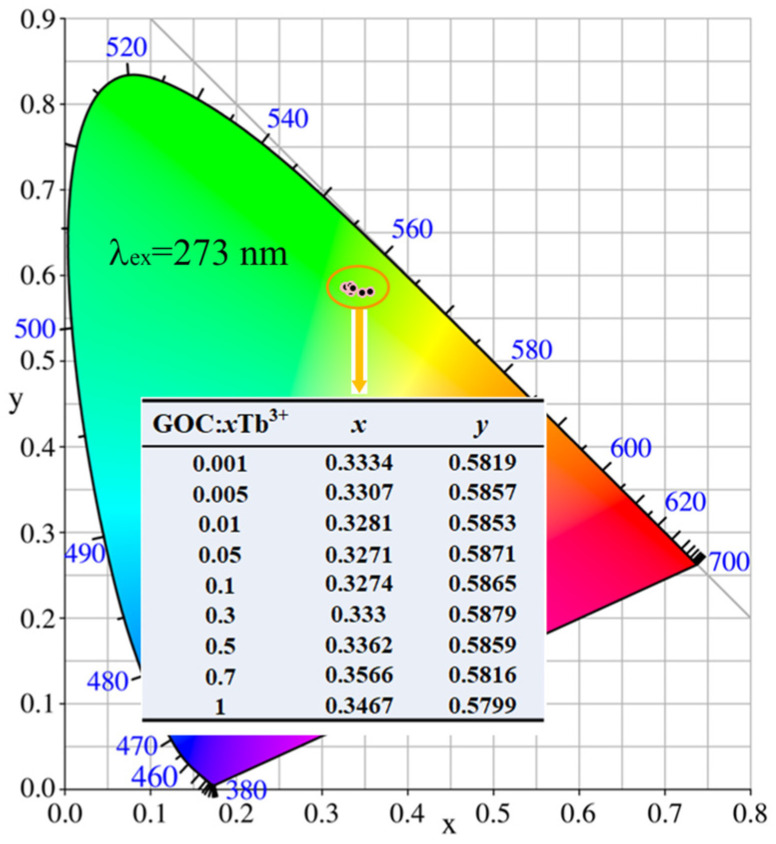
Chromaticity coordinates of GOC:*x*Tb^3+^ excitation at 273 nm.

**Figure 11 molecules-29-03846-f011:**
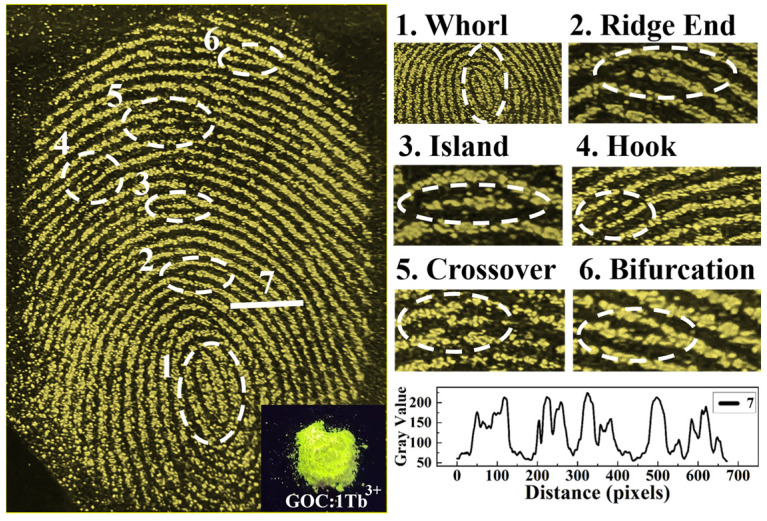
The latent fingerprint details and gray value of forefinger.

**Table 1 molecules-29-03846-t001:** Quantum yields of GOC:*x*Tb^3+^ under three excitation wavelengths.

x	0.001	0.005	0.01	0.05	0.1	0.3	0.5	0.7	1
λ_ex_ = 273 nm	15.33%	36.99%	55.60%	57.50%	44.25%	37.37%	27.96%	22.45%	14.11%
λ_ex_ = 311 nm	3.34%	17.35%	40.90%	42.77%	23.38%	18.04%	13.56%	7.51%	2.49%
λ_ex_ = 365 nm	~0	0.19%	3.38%	16.43%	17.66%	25.85%	29.65%	31.01%	33.73%

## Data Availability

The original contributions presented in the study are included in the article, further inquiries can be directed to the corresponding authors.

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
