# Peer review of "A Novel Synthesis Method of Dumbbell-like (Gd1−xTbx)2O(CO3)2·H2O Phosphor for Latent Fingerprint"

_molecules, 2024, doi:10.3390/molecules29163846_

Round 1

Reviewer 1 Report

Comments and Suggestions for Authors

The paper presents extensive and well-structured experimental data on a novel method for preparing of GOC:xTb3+ fluorescent powder. Considering the simplification of material synthesis conditions and its potential applications, such as protection against counterfeiting and concealed fingerprint visualization, the article is valuable and provides relevant content.

However, I would like to note the following:

1. The abbreviation "GOC" should be defined upon its first mention.

In section 2.4 (first paragraph), the statement "The results indicate that the excited state electrons of Tb3+ are generated through two processes: stimulated transitions and resonant energy transfer" requires further justification.

Author Response

Comments 1: [The abbreviation "GOC" should be defined upon its first mention.]

Response 1: Thank you to the expert for recognizing and supporting our work. The abbreviation GOC has been defined in the text.

    “In the biomedical field, the excellent magnetic and optical properties of Gd2O(CO3)2·H2O (GOC) have been applied separately in magnetic resonance imaging [6, 7] and luminescence[8, 9].”

Comments 2: [In section 2.4 (first paragraph), the statement "The results indicate that the excited state electrons of Tb3+ are generated through two processes: stimulated transitions and resonant energy transfer" requires further justification.]

Response 2: Thank you to the expert for pointing out the incomplete expressions in the article. In Section 2.4, the excitation spectrum of GOC:1Tb3+ is shown in Figure 6. In GOC:1Tb, Tb3+ completely replaces Gd3+, and its chemical formula is Tb2O(CO3)2·H2O. In the excitation spectrum of GOC:1Tb, it can be observed (orange line) that Tb3+ ions can be well excited at 254, 273 and 284 nm, exhibiting a "stimulated transition" phenomenon.

  When the fluorescent powder contains both Tb3+ and Gd3+ ions, we found that the excitation intensity at 273 nm is significantly enhanced. Based on the band overlap of Gd3+ and Tb3+ ions at this peak, we conclude that excitation enhancement based on resonance energy transfer occurred simultaneously at 273 nm. Therefore, we have modified the corresponding sentences in the text to make the expression more precise.

    “The results indicate that the excited state electrons of Tb3+ are generated through two processes: stimulated transitions and resonant energy transfer, when it is excited by 273 nm wavelength.”

Reviewer 2 Report

Comments and Suggestions for Authors

1) Main Question Addressed by the Research:
The primary question addressed by the research is how to develop a more efficient and simplified method for synthesizing GOC phosphors using sodium carbonate, and whether this method can achieve stable chemical composition and optimal fluorescence properties suitable for applications such as latent fingerprint visualization.

2) Originality: The novel use of sodium carbonate for synthesizing GOC phosphors, as opposed to traditional urea methods, is original. This new method simplifies the synthesis process and stabilizes the chemical composition.
Relevance: The study addresses the specific gap in the field related to the stringent preparation conditions of the urea method, offering a practical alternative.
Gap Addressed: The paper specifically addresses the need for a simpler, more stable, and scalable synthesis method for high-performance phosphors with potential applications in fluorescence recognition and fingerprint visualization.

3)  This research adds a novel synthesis methodology using sodium carbonate, which is less dependent on stringent reaction conditions compared to the urea method. The study also provides a detailed analysis of optimal conditions for achieving high fluorescence intensity and quantum efficiency, offering practical insights for the mass production of these materials.

4) Specific Improvements for Methodology:
No improvements required

5) The conclusions are consistent with the evidence presented in the abstract and conclusions sections. The study systematically shows that the sodium carbonate method produces stable and efficient phosphors under optimal conditions. All main questions posed, including the efficacy of the new method and its practical applications in fingerprint visualization, were addressed through specific experiments involving elemental mapping, fluorescence intensity analysis, and quantum efficiency measurements.

6) The references appear appropriate and relevant to the study. However, attention should be given to correcting any formatting issues, such as highlighted parts in references 30 and 35.

7) No improvements needed for Tables, Figures, and Data
---------
See attached

Comments on the Quality of English Language

Minor revisions required

Author Response

Comments 1: [Even though it can be probably inferred from the context, I feel that the acronym “GOC” should be explicitly defined at the beginning of the article.]

Response 1: Thank you to the expert for recognizing and supporting our work. The abbreviation GOC has been defined in the text.

“In the biomedical field, the excellent magnetic and optical properties of Gd2O(CO3)2·H2O (GOC) have been applied separately in magnetic resonance imaging [6, 7] and luminescence[8, 9].”

Comments 2: [Line 62:”In the chemical reactions at a constant pH value of 9.5, the amount of Na2CO3 increases”. Maybe you mean that the amount “has been increased”?]

Response 2: Thank you for pointing out the inappropriate expressions in the article, we have revised the sentence.

“In the chemical reactions at a constant pH value of 9.5, the amount of Na2CO3 has been increased from 3 to 11 mmol, as shown in Figure 1a.”

Comments 3: [Line 118: “An axial grain handle connects the mushroom heads at both ends. An axial grain handle connects the two ends.”. I think that there is a repetition in this sentence.]

Response 3: Thank you for pointing this out. We have deleted the redundant sentences in the article.

Comments 4: [Line 169-170:“For Tb3+ doping levels between 0.005-0.3, excitation at 273 and 311 nm are effective. For doping levels above 0.3, excitation at 365 nm is more effective”. Can the authors try to rationalize or propose an explanation for this finding?]

Response 4: Thank you for your constructive suggestions. We have added corresponding explanations in the text.

“For Tb3+ doping levels between 0.005-0.3, excitation at 273 and 311 nm is effective. This is primarily attributed to the high content of Gd3+ in GOC:xTb3+, where Gd3+ ions transfer excited state photons to Tb3+ through energy resonance. When the Tb3+ content exceeds 0.3, the intensity of f-f transitions of Tb3+ ions is proportional to the Tb3+ content, and excitation at 365 nm becomes more effective.”

Comments 5: [Line 275:“In the experimental section” maybe “In the experimental procedure”.]

Response 5: Thank you for your guidance. We have corrected our expression.

Comments 6: [In the references section, ref 30 and 35 appear to have some parts highlighted in yellow.]

Response 6: Thank you for your reminder. We have carefully checked the references

Reviewer 3 Report

Comments and Suggestions for Authors

This manuscript presents a novel synthesis method for phosphors, which is comprehensively demonstrated. It is recommended that this be published in this journal after a few minor revisions.

1. In the Abstract, the dosage of Na2CO3 should be shown in quantitative amounts, not as a number of times.

2. In the last paragraph of the Introduction, the sentence 'a novel method for preparing sodium carbonate was proposed' is not in context; please revise it.

3. In section 2.1, the phrases 'at a constant pH value of 9.5' and 'at a constant Na2CO3 dosage of 7 mmol' should be inserted into the caption of Fig. 1.

Author Response

Comments 1: [In the Abstract, the dosage of Na2CO3 should be shown in quantitative amounts, not as a number of times.]

Response 1:Thank you for your suggestion. The description has been revised.

Original:(line 11-12)

“Stable chemical composition of phosphor can be obtained when the dosage of Na2CO3 and the pH value of the solution are 3-11 times and 8.5-10.5, respectively.”

Revised:

“1 mmol of stable fluorescent powder can be widely prepared using 3-11 mmol of Na2CO3 at a pH value of 8.5-10.5 in the reaction solution.”

Comments 2: [In the last paragraph of the Introduction, the sentence 'a novel method for preparing sodium carbonate was proposed' is not in context; please revise it.]

Response 2: The expression has been revised as follows:

Original: (line 50)

“In this work, a novel method for preparing sodium carbonate was proposed.”

Revised:

In this work, a novel method for preparing GOC:xTb3+ using sodium carbonate was proposed.

Comments 3: [In section 2.1, the phrases 'at a constant pH value of 9.5' and 'at a constant Na2CO3 dosage of 7 mmol' should be inserted into the caption of Fig. 1.]

Response 3: Thank you for your reminder. We have added the corresponding description

Original: (line 77-78)

Figure 1. XRD patterns of GOC:0.05Tb3+ prepared at different (a) contents of Na2CO3; (b) pH values.”

Revised:

Figure 1. XRD patterns of GOC:0.05Tb3+ prepared at different (a) contents of Na2CO3 at a constant pH value of 9.5; (b) pH values at a constant Na2CO3 dosage of 7 mmol.”